# Vitamin D3 Loaded Niosomes and Transfersomes Produced by Ethanol Injection Method: Identification of the Critical Preparation Step for Size Control

**DOI:** 10.3390/foods9101367

**Published:** 2020-09-26

**Authors:** Oscar R. Estupiñan, Pablo Garcia-Manrique, Maria del Carmen Blanco-Lopez, Maria Matos, Gemma Gutiérrez

**Affiliations:** 1Instituto de Investigación Sanitaria del Principado de Asturias (ISPA), Hospital Universitario Central de Asturias, 33006 Oviedo, Spain; UO217649@uniovi.es; 2CIBER en Oncología (CIBERONC), 28029 Madrid, Spain; 3Department of Physical and Analytical Chemistry, University of Oviedo, 33006 Oviedo, Spain; garciampablo@uniovi.es (P.G.-M.); cblanco@uniovi.es (M.d.C.B.-L.); 4Department of Chemical Engineering and Environmental Technology, University of Oviedo, 33006 Oviedo, Spain; matosmaria@uniovi.es; 5Asturias University Institute of Biotechnology, University of Oviedo, 33006 Oviedo, Spain

**Keywords:** niosomes, transfersomes, surface tension, ethanol injection method, encapsulation efficiency, vesicles stability

## Abstract

Vesicular nanocarriers have an important role in drug delivery and dietary supplements. Size control and optimization of encapsulation efficiency (EE) should be optimized for those applications. In this work, we report on the identification of the crucial step (injection, evaporation, or sonication) innanovesicles (transfersomes and niosomes) preparation by theethanol injection method (EI). The identification of each production step on the final vesicle size was analyzed in order to optimize further scale-up process. Results indicated that the final size of transfersomeswas clearly influenced by the sonication step while the final size of niosomes was mainly governed by the injection step. Measurements of final surface tension of the different vesicular systems prepared indicate a linear positive tendency with the vesicle size formed. This relation could help to better understand the process and design a vesicular size prediction model for EI. Vitamin D3 (VitD3) was encapsulated in the systems formulated with encapsulation efficiencies larger than 90%. Interaction between the encapsulated compound and the membrane layer components is crucial for vesicle stability. This work has an impact on the scaling-up production of vesicles for further food science applications.

## 1. Introduction

The use of vesicular systems as nanocarriers for bioactive compounds has significantly increased in recent years. Physico-chemical features of these systems allow to create easy-to-produce nano-scaled transporters for drug delivery or dietary supplements. They are able to encapsulate either polar compounds, in the inner aqueous compartment of the vesicle, or non-polar molecules, embedded in the membrane.

Vitamin D3 (VitD3) is a hydrophobic molecule required in the human diet to maintain good health and well-being [1]. VitD3 deficiency is frequently found in people who are not exposed enough to sunlight or with any gastrointestinal disorder.

Calcitriol is the biological form of VitD3, which plays a critical role in osteoporosis prevention, enhancing calcium absorption on the human metabolism [2,3].

VitD3 is sensitive to environmental factors since it can be easily oxidized. This leads to the loss of its functionality [1]. For these reasons, VitD3 is often encapsulated within colloidal delivery systems, such as nanoparticles, nanovesicles, nanoemulsions, or macroemulsions [2,4,5,6], which will protect it and even control its release [7].

Nature and size of the nanovesicles are crucial parameters related to the capacity to avoid the immune system, or trespassing physiological barriers [8,9]. Therefore, control of nanovesicles characteristics should be achieved in order to develop important applications for biomedicine, [10,11,12] cosmetics, and food industries [13,14,15,16,17,18,19,20,21,22,23]. This is also important in order to develop new strategies for the synthesis of novel nanomaterials [24,25,26].

Depending on its membrane composition, main artificial vesicular systems are classified as liposomes, niosomes, or transfersomes. Liposomes are composed of phospholipids, which allows the production of small vesicles. By comparison, niosomes are composed of non-ionic surfactants that produce a more stable, less toxic, and more flexible vesicles [27,28]. Transfersomes include both phospholipids and surfactants, sharing features of both types of systems. Cholesterol (Cho) is commonly used as a membrane additive for niosomes preparation to improve the stability of vesicles, membrane elasticity, entrapment efficiency, and controlled release [27]. However, cholesterol slightly increases vesicle size. It is reported that cholesterol also plays a fundamental role in niosomes formulation when hydrophilic surfactants (hydrophilic-lipophilic balance, HLB around 10) are used as main membrane compounds [11].

Vesicle size is governed by composition and the selected preparation method. The conventional preparation methods are thin film hydration, reverse-phase evaporation, solvent (frequently ethanol) injection, and ethanol injection method (EI) [29,30,31,32]. Other more recent preparation methods are described for liposomes formation such as detergent deplection,but their suitability for transfersomes and niosomes preparation has not been widely explored yet [33]. Other techniques such as microfluidics [34]offer good control fornanovesicles preparation, but their scale-up is more complicated due to the specific and delicate material requiredfor the microchannels. These could be easily operated by precipitation of some components,and hence, easily damaged. Moeover, the use of high temperatures required for surfactants self-assembly is not eailsy achieved. EI is an easy set-up method with high flexibility to scale-up the batch production of nanovesicles [31]. This method allows obtaining small unilamellar vesicles with a simple set up based on the injection of an ethanolic membrane compound solution in an aqueous phase and further evaporation of the organic solvent. Although this method forms vesicles without the use of further agitation/sonication step, as it was firstly reported by Batzri and Korn [35], very frequent sonication is used in order to reduce vesicle size [8]. There are recent works on the characterization of the vesicles obtained by the EI method, even at the large scale [31]. In addition, the effect of several operating parameters on the final vesicle size has been reported [36]. However, it is also important to identify the individual role of each step (injection, evaporation, and sonication) on the final vesicle size for formulations with and without phospholipids on its membrane layer. This is crucial for a well particle size control for each type of vesicle and scaling-up.

In this paper, we have identified the crucial step for size control and scaling-up production of niosomes and transfersomes prepared by the EI method. With this purpose, the effect of membrane composition on each step was studied, together with the influence on average size, polidespersity index (PDI), VitD3 encapsulation efficiency (EE), and stability of the prepared systems. We have found that surface tension measurements could be correlated to particle size and, to our knowledge, this has not been reported before in the literature. This could have an impact on the development of mathematical models to optimize nanovesicle size during the injection step,and to predict the effect of membrane components for scaling-up the production of vesicular nanocarriers for further food industry applications.

## 2. Materials and Methods

### 2.1. Vesicles Formulation

Membrane components were selected taking into account their biocompatibility, hydrophilic-lipophilic balance (HLB), and its critical packing parameter. Phospholipid component was Phospholipon 90G^®^supplied by Lipoid, Ludwigshafen Germany (PC, C_42_H_80_NO_8_P, molecular weight (MW = 775.04 g/mol). Selected nonionic surfactants were Span^®^ 60 (Sp60, C_24_H_46_O_6_, MW = 430.62 g/mol, HLB = 4.7), Tween^®^20 (Tw20, C_58_H_114_O_26_, MW = 1227.54 g/mol, HLB = 16.7), and Tween^®^ 80 (C_64_H_124_O_26_, MW = 1310 g/mol, HLB = 15.0); all of them supplied by Sigma-Aldrich/Merck, USA. Finally, cholesterol (Cho, C_27_H_46_O, MW = 386.65 g/mol) supplied by Thermo Fisher Scientific, USA was added to the formulations as membrane stabilizer. VitD3 for required systems was supplied by Sigma-Aldrich, San Luis, MO, USA.

### 2.2. Preparation Method

Nanovesicles were prepared by the EI method using optimized production parameters detailed in previous works [37]. Three different steps were followed for nanovesicles formation by the EI method: (i) Membrane compounds (PC, non-ionic surfactants and /or cholesterol) were dissolved in absolute ethanol and injected at 60 °C in a heated phosphate buffer salinity (PBS) which was also at 60 °C and gently stirred at 400 rpm with a magnetic stirrer. The injection was made at a constant flow rate of 130 mL/hby the use of a syringe pump. In all cases, 5.6 mL of organic phasewereinjected into a 50 mL of aqueous phase (PBS). (ii) Organic phase was mostly eliminated (up to the azeotropic point) using a rotary evaporator working at 80 mbar in a heating bath at 50 °C. (iii) Finally, nanovesicles prepared were sonicated by using an ultrasonic probe for 20 min at a wave amplitude of 55% and power energy of 900 watts. During this process, samples were ice cooled to avoid excessive heat of sonication. The three steps process indicated is graphically described in Figure 1.

All nanovesicles formulated are summarized in Table 1, where the ratio of used composition and final particles sized obtained are summarized.

### 2.3. Vitamine D3 Encapsulation

Vit D3, a lipophilic molecule (C_27_H_44_O, MW = 384.64 g/mol), was added to the organic phase at a concentration that ensures that the final sample will contain 0.02 mg/mLof it. The molecular structure of Vit D3 is presented in Figure 2.

### 2.4. Characterization of Nanovesicles

Nanovesicles size and polydispersity index (PDI) were characterized using Malvern Zetasizer Nano ZS (Malvern Instruments Ltd., Malvern, UK) by Dynamic Light Scattering (DLS).

Zeta-potential of all samples were analyzed by Malvern Zetasizer Nano ZS (Malvern Instruments Ltd., Malvern, UK) using Dopplerelectrophoresis laservelocimetry.

The stability of nanovesicles was analyzed by multiple light scattering (MLS) with a Turbiscan Lab Expert equipment (Formulaction, France). The instrument operates by sending a light beam through a cylindrical glass cell containing the sample. The light source is an electro luminescent diode in the near infrared with a wavelength of 880 nm. Part of the incident light is then backscattered by the sample or transmitted through it, and received by two sensors with different location: the sensor that receives the transmitted light is located 180° from the incident radiation, while the sensor that receives the backscattered light is located 45° from the incident light. Samples were placed in the test cells, and the transmitted/backscattered light was monitored as a function of time and cell height for 8 days at 30 °C using the Ageing Station (Formulaction, l’Union, Toulouse, France). The profiles build up a macroscopic fingerprint of the sample at a given time, providing useful information about changes in droplet size distribution, or appearance of a creaming layer, or a clarification front with time. The Turbiscan stability index (TSI) was also obtained to compare the stability of the different formulations studied. It sums all the variations detected in the samples in terms of size and/or concentration in the studied period of time, and is defined by the following Equation (1):(1)$$\mathrm{TSI}=\sum_{i}\frac{\sum_{i}\left|{\mathrm{scan}}_{i}-{\mathrm{scan}}_{i-1}\right|}{H},$$
where i is the scan number and H is the height of the cell where the samples was placed.

Superficial tension of the samples was determined with KSV sigma 700 (KSV Instruments, Stockport, Sweden) tensiometer at constant temperature of 20 °C by the ring method.

EE was calculated as the ratio between the quantity of encapsulated compound (after proper purification), and the total amount in the unpurified suspension according to Equation (2):(2)$$EE=\;\frac{{\left[compound\right]}_{encapsulated}}{{\left[compound\right]}_{initial}}\times 100.$$

The quantification of the cargo molecules was carried out by RP-HPLC (HP series 1100 chromatograph, Hewlett Packard, Agilent Technologies), with a Zorbax Eclipse Plus C18 column (4.6 mm × 150 mm, 5 μm, Agilent Technologies, Santa Clara, CA, USA).UV/vis (HP G1315Adetector, Agilent Technologies) and fluorescence (1260 Infinity A detector, from Agilent Technologies), were used as detection coupled to the chromatographic separation. As mobile phase, a miliQ water/methanol gradient was used at a caudal of 0.8 mL/min.

Before performing RP-HPLC analysis to determine encapsulation efficiency (EE), the non-encapsulated compound had to be removed by passing the sample through a Sephadex G-25 packed column (GE Healthcare Life Sciences, Pittsburgh,PA, USA) at a constant flow rate of 200 mL/h. Then both samples, filtered and non-filtered, were diluted 1:10 (*v*/*v*) with methanol to facilitate vesicle rupture and to extract the encapsulated compound.

### 2.5. Statistics Analysis

Data were expressed as the mean ± SD (standard deviation) of three independent experiments, and statistical analysis of the data was carried out (ANOVA). Fisher’s test (*p* < 0.05) was used to calculate the least significance difference (LSD) using statistical software Microsoft Excel.

## 3. Results

### 3.1. Effect on Particle Size

The amount of membrane compounds added to the organic phase was optimized using Span^®^60 and Cho 1:1 as membrane components.This formulation was used since it is one of the most common formulations studied in previous works [37,38]. The studied range was changed from 2.2 to 8.3 (%*w*/*w*). It was observed that when the mass increases from 2.2 to 4.3%*w*/*w*, the final particle size of the formulated nanovesicles increased from 209 to 248 nm, while a significant difference (*p* < 0.05) was not observed when the mass of the membrane compounds increased to a higher value (from 4.3 to 8.3%*w*/*w*). Moreover, a larger concentration of membrane compounds resulted in increasedPDI values. Results are depicted in Table 1, for samples A1, A2, A3, and A4.

Multiple formulations of niosomes and transfersomes, based on previous studies [13,37,38], were analyzed with a constant mass concentration in the organic phase(2.2%). Small sizes are more interesting from a food, pharmaceutical, and biomedical point of view, since small vesicles can be easily absorbed by human cells. Results of the final particle size and PDI are presented in Table 1 for all the niosomes and transfersomes prepared. Samples A1, B, and C corresponded to three different niosomal formulations, while samples D, E, F, and G corresponded to transfersomes.

In the case of niosomes, it can be observed that Cho was partially replaced by non-ionic surfactant with high water solubility, such as Tween^®^20 and Tween^®^80 (samples B and C). It is known that larger amounts of Cho are more advantageous because they increase membrane stability and resistance [27]. However, the combination of non-ionic surfactants with high and low HLB values as membrane compounds has been found to be an appropriate selection for higher size reduction due to the reduction of its critical packing parameter (CPP). Moreover, higher encapsulation efficiencies (especially for encapsulation of hydrophilic molecules) were found by the use of high water solubility non-ionic surfactants [11,30]. Significant size reduction was observed by the use of Tw20 as a partial replacer of Cho, (from 209 to 186 nm), while with the use of Tw80, no significant size reduction was observed. Moreover, the use of Tw20 as a partial replacer of Cho results in a lower PDI value, indicating a more monodisperse sample formulation, which is not the case when Tw80 was used. The double bond found in the hydrocarbon chain of Tw80 surfactant could be responsible for a slight expansion of membrane compounds versus the use of Tw20 [39].

For transfersomes formulations, several combinations of PC, Sp60, Tw20, and Cho were selected. Tw80 was not used due to itsnegative effect on niosomes formulation (samples D, E, F, and G). Looking at the results obtained by the formulation of transfersomes, it could be appreciated that the use of PC significantly reduced the main size of the nanovesicles, since all transfersomes presenteda lower size than the prepared niosomes. The smallest size (88 ± 5.4 nm) was obtained by the sample with a higher amount of PC and without the presence of Cho, with a significant lower value than the other transfersomes prepared (*p* < 0.05). This corroboratesonce again the nanovesicle membrane layer expansion.

For the determination of the effect of each step, particle size was measured after each preparation step (injection, evaporation, and sonication). Theresults are presented in Figure 3A for niosomes and Figure 3B for transfersomes. Some samples did not present significant size differences (*p* > 0.05) between them, and therefore they have been labelled with the same letter as Figure 3.

It is also important to point out that all formulations are expected to produce unilamellar vesicles since EI method tends to produce this type of nanovesciles with relatively small final size compared to other types of methods such as the thin film hydration method, which has been reported to produce large multilamellar vesicles [40].

Significant size reduction (*p* < 0.05) was obtained after each step in all cases. For both types of tested formulations, it can be observed that the determinant step for vesicles formation by the EI method was the injection step, so the optimization of the operating conditions during the injection is crucial in order to control the vesicle size.

The evaporation step, in which most of the ethanol is eliminated, reduces 11–18% of the sizes with respect to the size obtained after the injection (for both types of formulations tested). This is probably due to the fact that the membrane components must be more packed to satisfy their lipophilic behavior once the ethanol is removed from the system.

Regarding the sonication step, a difference was found in both types of systems, i.e., niosomes size was reduced around 7–18% during the sonication step. A larger significant (*p* < 0.05) reduction in size of all the cases was observed for the formulation containing non-ionic surfactants with high HLB value (sample B and C). However, for transfersomes, the values reduction rose in the range of 50–60%, significant in all cases (*p* < 0.05). This difference could be explained because of the higher fluidity and elasticity defining the transferosomes membrane in comparison to niosomes [39].

This observation indicates that sonication is a crucial step for transfersomes formation with the EI method, and hence optimization of the operation parameters in that step is necessary in order to control the final nanovesicle size. It would bereasonable to attribute this different behavior between both types of samples to the presence of PC, since is the component that makes the difference between both types of formulations. After sonication, all transfersomes presented similar values of the particle size. However, high differences were found after injection.

In the case of transfersomes formulations, the largest size observed after injection was obtained for sample G. It is important to take into account that this sample is the one that contains a higher portion of Sp60 and one of the less portion of PC. This makes its self-assembling behavior more similar to the one observed by niosomes, but the content of PC could be related to the crucial role of the sonication step at final size determination.

When usual amphiphilic molecules used for membrane vesicles formation are dissolved in a solution, they tend to be located at the air–liquid interface in order to satisfy their amphiphilic character. Depending on the type of membrane compounds used and the concentration and the combination of the membrane compounds selected, several types of colloidal systems can be found in the bulk solution, withvesicles and micelles being the more common ones [40,41]. Referring to an aqueous solution in which membrane compounds are added, this can be summarizedas: high lipophilic character molecules will produce vesicles (or nanovesicles), while high hydrophilic molecules will produce micelles. When a small portion of the hydrophilic molecule is added to a solution that contains hydrophilic molecules, vesicles can be formed using the combination of all added membrane compounds. However, depending on the concentration and ratio between used compounds, micelles and vesicles can be found simultaneously [40].

Since injection of the organic phase into the aqueous phase seems to be a determining step for both types of vesicles studied in the present study (niosomes and transfersomes), surface tension of prepared samples after the injection step was measured in order to study a possible influence of this parameter in the final size obtained after this predominant step (Figure 4). To avoid the effect of the particular membrane compounds concentration,only the values with 2.2 *w*/*w* of the membrane compounds concentration on the organic phase were presented in Figure 4.

A positive relation was observed between both variables presented. It was found that the surface tension of a sample was reduced due to the presence of the amphiphilic molecules at the interface (liquid–air). The lower the surface tension, the higher the packing of the molecules at the liquid–air interface [40,42,43]. The more compact the molecules were, the smaller the curvature radius they adopt when theyare forming the nanovesicles membrane. Therefore, smaller nanovesicleswould beexpected.

Figure 4 shows how amphiphilic molecules wouldbe disposed at the liquid–air interface and how they wouldbe disposed to form nanovesicles, using the same amount of molecules and high and low molecules packing.

### 3.2. Vesicles as Biocompounds Carriers

The effect of composition and final size on the biocompound encapsulation was studied. For this purpose, VitD3 was selected. It is known that this compound could yield high encapsulation efficiency in vesicles,due to its lipophilic character [38]. In this set of experiments, a total weight mass of 2.2% was used for all prepared vesicular systems.

The presence of VitD3 on the formulated systems increasedthe size of the nanovesicles between 12–20 nm for all formulated systems. This is expected since the VitD3 molecules encapsulated wouldbe trapped at the membrane nanovesicles, reducing the membrane molecules packing and increasing the final nanovesicle sizes. Similar results were obtained in recent works withdifferent types of drugs encapsulated intoniosomes formulated with different hydration media by the thin film hydration method [30]. It has been found that the encapsulated drug and the compounds used in the hydration media could significantly change the membrane nanovesicle disposition, and hence, increase their final size. However, it is expected that the encapsulation of hydrophilic compounds, which are incorporated into the membrane core instead of the membrane layer wouldhave more influence on the final nanovesicle size.

#### 3.2.1. Encapsulation Efficiency

Due to the high lipophilic character of VitD3, their encapsulation will take place on the vesicle membrane. EE of VitD3 in niosomes and transfersomes are presented in Figure 5.

It can be observed that EE of VitD3 in both types of systems has registered values between 87.6 and 98.2%. Even though no significant differences (*p* > 0.05) were observed between all systems tested, some trends can be withdrawn. Even all values obtained were high, the general trend indicated that the EE was larger on transfersomes than in niosomes. Looking at the results obtained by the encapsulation into niosomal systems, it can be observed that the presence of high HLB value surfactants increased the EE of VitD3. This could be related to the higher packing of the membrane vesicle when Tw20 surfactant was present, which could increase the retention of the VitD3 molecule. Regarding the results obtained by the formulated transfersomes, it can be appreciated that formulations E and F present the high EE, which correspond to the formulations with a higher content of PC. These again correspond to the systems with higher packing at the membrane layer, and those with smaller nanovesicle size.

#### 3.2.2. Stability

Stability of all samples formulated with and without VitD3 encapsulated was monitored for a period of time of 8 days. Turbiscan stability index (TSI) was analyzed for all samples formulated and is presented in Figure 6, niosomes (Figure 6A) and trasnfersomes (Figure 6B).

As a general trend, after 8 days of analysis, nisomes presented higher stability (lower TSI value) than transfersomes, and amaximum TSI value of 18 was registered for niosomes while transfersomes show TSI values of around 15. Regarding niosomes (Figure 6A), a clear dependence of stability with niosomalformulation was found, being less important the effect of the encapsulated VitD3 over the membrane nanovesicle. Very similar TSI values were obtained for each formulation with and without the encapsulated drug. Higher stability was observed for niosomes with no presence of Tween^®^surfactants on the membrane layer, and even lower for those niosomes that contained surfactant Tw80 on the nanovesicle membrane layer (sample C). This trend can be explained by the fact that the lower hydrophobicity of the membrane vesicle could enhance the membrane layer vesicles widening. Some aqueous media could be going through it and reach the internal aqueous core, which also could have a negative effect on the EE. Looking at the particular effect of the presence of VitD3 at the niosomes membrane layer, it could be appreciated that the TSI value was slightly higher than the value for empty nanovesciles. This can be attributed to the fact that the VitD3 molecule also has double bonds at its molecular structure (Figure 2), which could produce an effect similar to that produced by the presence of Tw80, thus increasing the nanovesicle membrane polarity.

Transfersomes formulations offered higher stability (lower TSI values) for those formulations which contain a lower amount of Tw20 surfactant (E and G), once again indicating that the slight increase of the polarity of the membrane layer decreases the satbilty of vesicles. Even higher stability (lower TSI value) was found for samples E which contains a higher amount of PC and loweramount of SP60 than samples G, increasing then the hydrophobicity character of the membrane bilayer.

Stabilities of samples D and F, the ones with a higher content of Tw20 surfactant, presented a clear difference between non-loaded and VitD3 loaded nanovesicles. The presence of VitD3 on the nanovesicle membrane layer seemed to have a negative effect on the vesicle stability, especially in the case of sample F. Stability of sample F was exactly the same for the system with and without encapsulated VitD3 on its membrane layer during the first four days, and from this point,the stability of the VitD3 loaded nanovesicle started exponentially to decrease. It is important to point out that sample F was the only one that does not present Cho on their formulation, which is known to be a membrane stabilizer, which indicates that the presence of Cho or other stabilizer compounds is especially important when nanovesicles were used for the encapsulation purpose.

Backscattering profiles of all samples with and without are presented in Figure 7 and Figure 8 (niosomes and transfersomes, respectivetly). Results confirmed that niosomes presented higher stability than tranfersomes as a general trend. In all transfersomesformulations, a precipitation layer was observed (Figure 8), since an increase on backscattering was obtained at the bottom part of the cell. For niosomes, this precipitation layer was nearly nonexistent, especially for sample A1 (Figure 7). Looking at the stability of niosomes (Figure 7), sample C presented the lower stability. Size reduction along time was observed due to the decrecrease on backscattering value, but no difference was observed for encapsulated and non-encapsulated samples for any of the niosomal formulations tested.

Looking at the transfersomes (Figure 8), it can be observed that sample E presented the highest stability, with just a slight increase in size along time (an increase of Backscattering value along time). Samples F and G present a decrease in size, indicating that some aqueous phase located in the nanovesicle core escaped to the external aqueous bulk, while the contrary effect was observed in sample D in which an increase in size was observed. The effect of the loaded drug was just observed in sample F, with higher instability when VitD3 was encapsulated. A decrease in size along time was analyzed with respectto the increase observed when the sample without the encapsulated drug was measured.

Zeta potential of all loaded and non-loaded samples was measured, but due to the non-ionic character of the surfactants used, all the systems presented lower negative values. Values between −2.8 and −5.4 mV were registered in all cases. However, sample A1 presented a higher absolute value, −18.5 and −21.5 mVfor empty and VitD3 encapsulated into the vesicular membrane, respectively. This samplecorresponded to the niosomal formulation with Sp60 and Cho, indicating that this formulation produced higher electrostatic repulsion than the other formulations tested.Similar zetapotential values were obtained in previous works when niosomes where prepared with Tween^®^ and Span^®^ surfactants [43]. It is important to point out that sample A1 was the one with higher stability (lower TSI value), but the value of the zeta potential registered could not be the single mechanism responsible for the high stability observed, since the value was not extremely high. The presence of VitD3 on the nanovesicle membrane layer introduced differences around −0.5 and −3.2 mV with respect to the empty systems.

## 4. Conclusions

The EI method is a good approach for the preparation of small unilamelar nanovesicles with a very simple set-up, being of special relevance,the operating conditions used during injection and sonication stages. The average size of the transfersomes’ vesicle was more dependent on the sonication step parameters, while the size of niosomes was clearly determined at the injection step. This indicates that the the use of the sonication step for niosomes preparation is not required. Control of the operating parameters of these steps should be crucial for the scale-up production of vesicular systems regarding thefinal size-dependent application.

A linear positive dependence between the size of nanovesicular systems and surface tension was found. This approachcould be useful for nanovesicles formulation since it could allow to predict the effect of the addition of membrane compounds onthe final size. This could also be helpful for mathematical model development, which could theoretically predict the final size of nanovesciles based on methods where the organic phase is injected into an aqueous phase.

Niosomes and transfersomes are suitable nanovesicular systems showing great potential as nano-scaled carriers for lipophilic compounds. The EE for VitD3 was higher in transfersomes thaninniosomes. Formulted systems contained between 2.0–2.4 mg/L of VitD3.

Stability of the nanovesicular systems was clearly influenced by the lipophilic character of the compounds present in the membrane bilayer formed. The widening or shrinking of the nanovesicles was observed in those systems where the content of either the membrane components or the encapsulated compounds increased the polarity of the membrane layer, enhancing the pathing of the aqueous external phase into the vesicles core. The presence of membrane stabilizers such as Cho plays a key roleto ensure nanovesicle stability along time. As a general trend, niosomespresentedahigher stability than transfersomes, either with or without encapsulated VitD3.

Bothformulated systems (either niosomesortranfersomes) could be suitable to be incorporated into food, cosmetic,or pharmaceutical products in order to supplement in a control manner a VitD3 providing health benefits on the human body.

## Figures and Tables

**Figure 1 foods-09-01367-f001:**
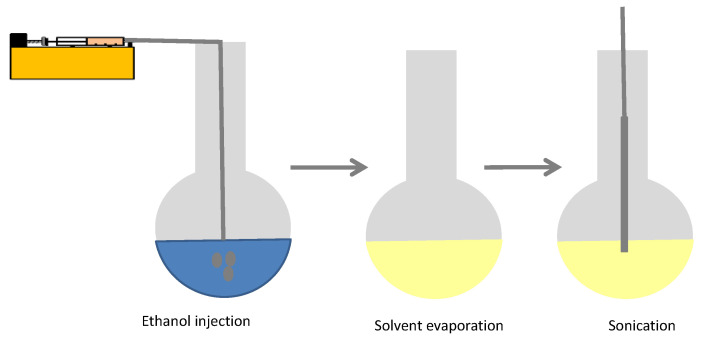
Schematic process of the ethanol injection method.

**Figure 2 foods-09-01367-f002:**
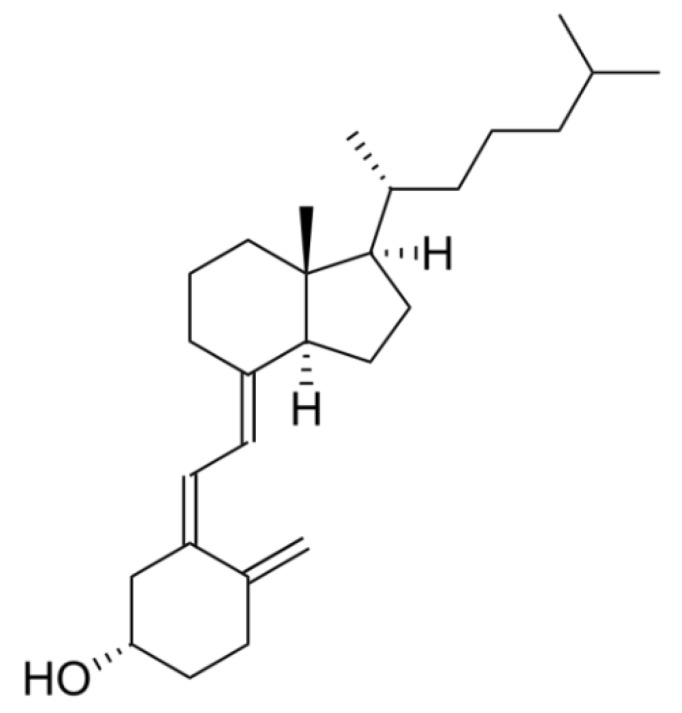
Molecular structure of Vitamin D3.

**Figure 3 foods-09-01367-f003:**
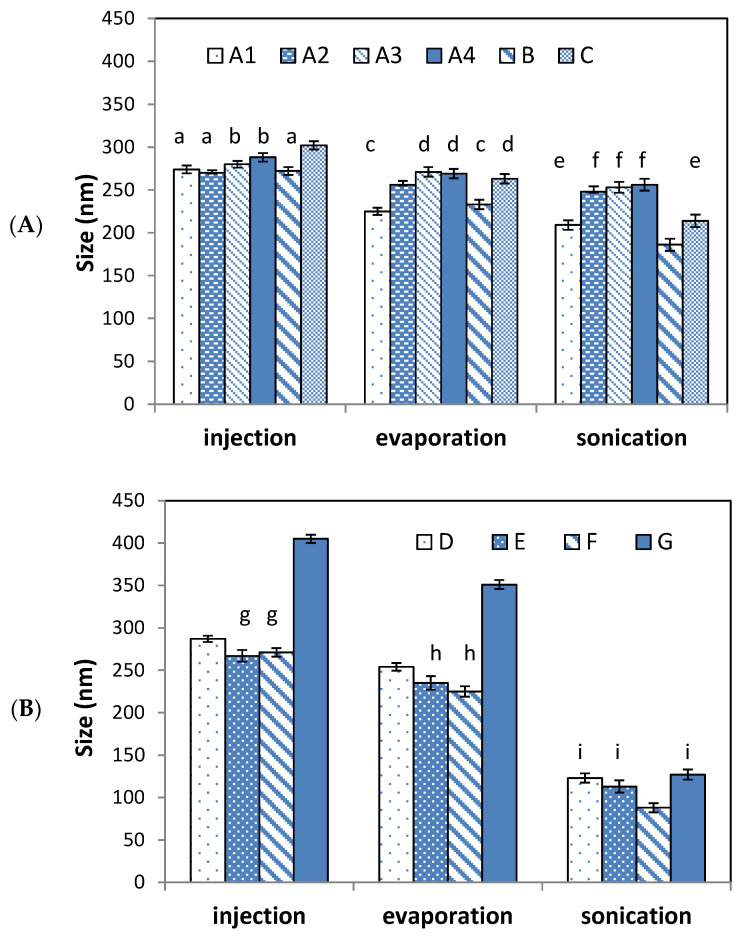
Mean particle size after each step of ethanol injection (EI) method for the preparation of niosomes (**A**) and transfersomes (**B**). The error bar corresponds to the standard deviation of triplicate samples. Figure shows significant differences (*p* < 0.05) after each preparation step for all samples. Same letters indicate samples without significant differences (*p* > 0.05) among them.

**Figure 4 foods-09-01367-f004:**
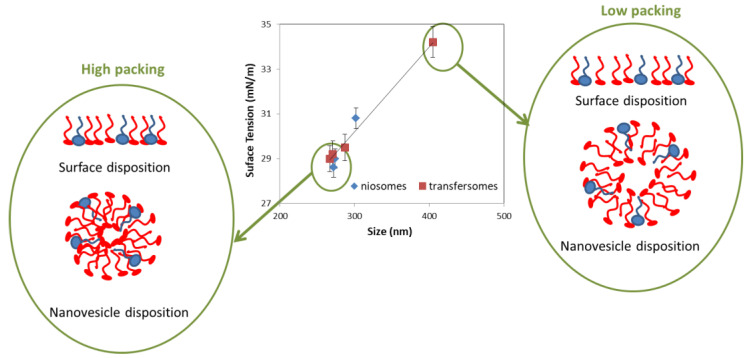
Surface tension versus vesicle size for systems formulated with 2.2%*w*/*w* membrane compounds in concentration on organic phase after the injection step. The error bar corresponds to the standard deviation of triplicate samples.

**Figure 5 foods-09-01367-f005:**
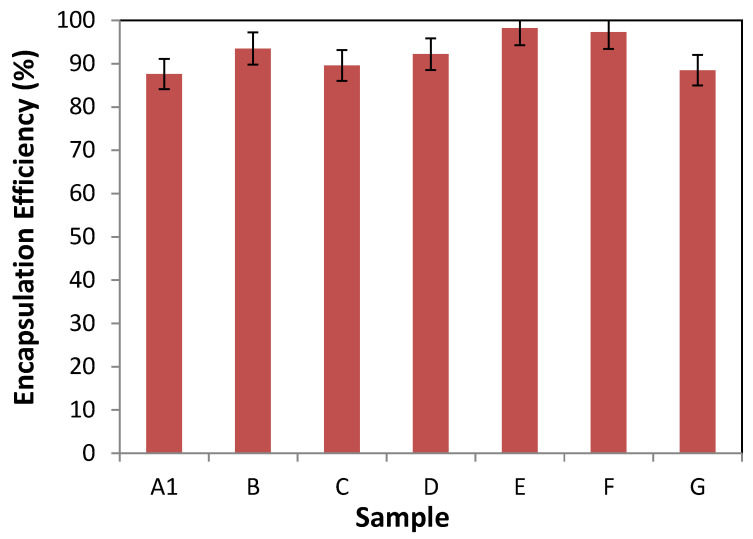
Encapsulation efficiency of vitamin D_3_ at several nanovesicular systems. Niosomes (samples A1, B, and C) and transfersomes (samples D, E, F, and G). The error bars correspond to the standard deviation of triplicate samples.

**Figure 6 foods-09-01367-f006:**
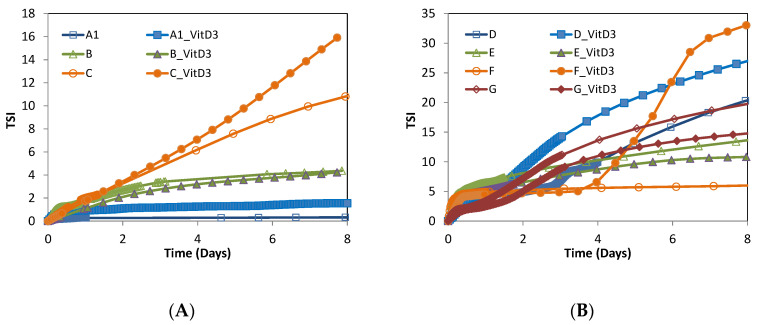
Turbiscan stability index (TSI) for vesicles encapsulating compounds, (**A**) niosomes and (**B**) transfersomes.

**Figure 7 foods-09-01367-f007:**
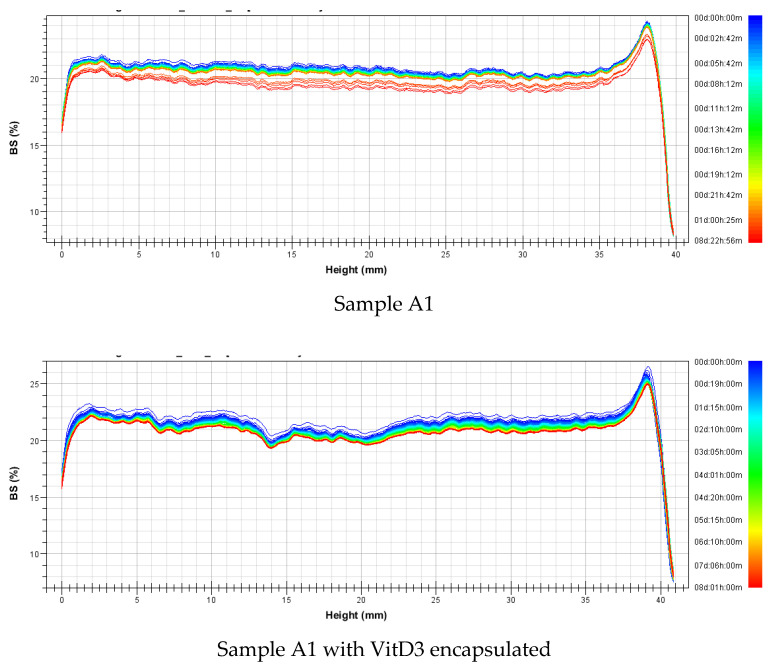
Backscattering profiles of niosomes formulated with and without VitD_3_ encapsulated.

**Figure 8 foods-09-01367-f008:**
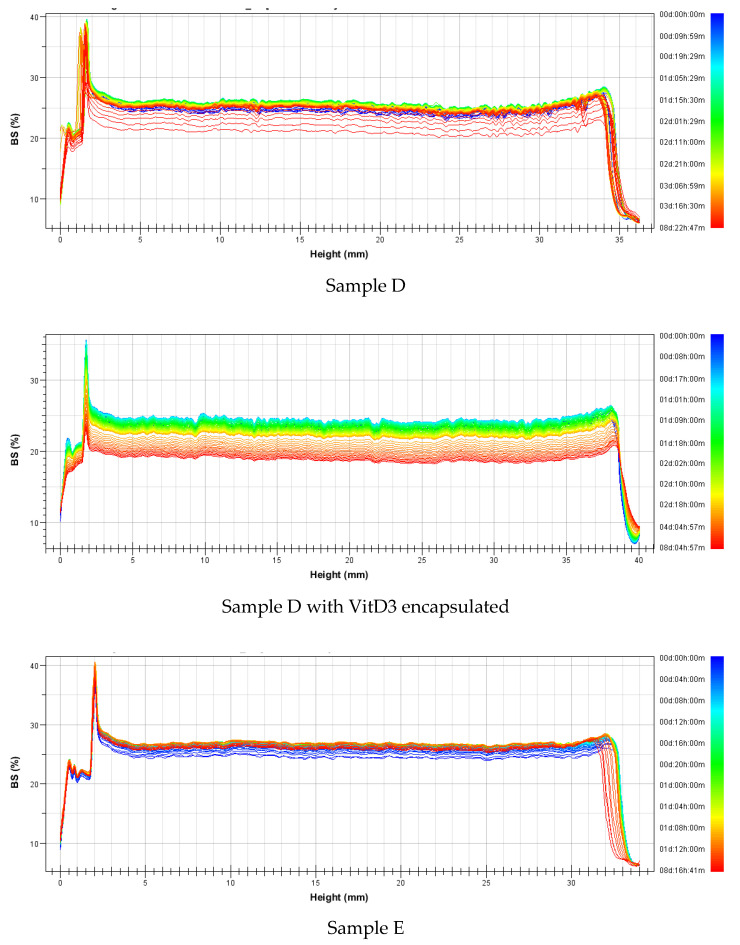
Backscattering profiles of transfersomes formulated with and without VitD_3_ encapsulated.

**Table 1 foods-09-01367-t001:** Niosmes and transfersomes particle size and polydispersity for several membrane formulations.

Nanovesicles	Sample	Mass (%w/w)	Formulation					Average Size (nm)	Polidispersity Index
			Span^®^60	Tween^®^20	Tween^®^80	Phospholipid	Cholesterol		
Niosome	A1	2.2	0.50	-	-	-	0.50	209 ± 4	0.18 ± 0.05
	A2	4.3	0.50	-	-	-	0.50	248 ± 5	0.17 ± 0.09
	A3	6.3	0.50	-	-	-	0.50	253 ± 4	0.227 ± 0.09
	A4	8.3	0.50	-	-	-	0.50	256 ± 5	0.2 ± 0.1
	B	2.2	0.50	0.25	-	-	0.25	186 ± 6	0.15 ± 0.01
	C	2.2	0.50	-	0.25	-	0.25	214 ± 5	0.161 ± 0.002
Transfersomes	D	2.2	0.13	0.29	-	0.29	0.29	123 ± 6	0.202 ± 0.008
	E	2.2	0.20	0.20	-	0.40	0.20	113 ± 7	0.21 ± 0.02
	F	2.2	0.25	0.25	-	0.50	-	88 ± 5	0.266 ± 0.001
	G	2.2	0.30	0.20	-	0.30	0.20	127 ± 5	0.199 ± 0.003

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
