# Peer review of "Vitamin D3 Loaded Niosomes and Transfersomes Produced by Ethanol Injection Method: Identification of the Critical Preparation Step for Size Control"

_foods, 2020, doi:10.3390/foods9101367_

Round 1

Reviewer 1 Report

O. Estupiñan and co-authors in this article titled "Vitamin D3 loaded niosomes and transfersomes produced by ethanol injection method: Identification of the critical preparation step for size control" claim to have identified for the first time "the identification of the crucial step at nanovesicles ( transfersomes and niosomes) preparation by Ethanol Injection Method ". In my opinion the work does not report elements of novelty with respect to the topic addressed and suffers from theoretical and methodological limitations, mainly the lack of a true experimental design.
Detailed comments follow.

Lines 62-63: “The conventional preparation methods are thin film hydration, reverse-phase evaporation and solvent (frequently ethanol) injection and ethanol injection method (EI)”.

Another established and fundamental method for vesicle preparation is the “Detergent Depletion”. An interesting and recent application in the field of drug delivery is found in "Preparation of drug-loaded small unilamellar liposomes and evaluation of their potential for the treatment of chronic respiratory diseases", DOI: 10.1016/j.ijpharm.2018.04.030.

Line 65: “This method allows to obtain small nanovesicles with narrow distribution”.

The liposomes prepared by solvent injection process are actually mostly polydispersed.

Line 69-70: “but the individual role  of each step (injection, evaporation and sonication) on vesicles size has never been reported before”.

An extensive literature reports that the  critical  process  parameters  that  can  affect  the vesicle  size  are  varied  lipids  proportion  in  ethanol, the rate of injection of lipids, magnetic stirring speed, and the phase volume ratio of solvent and nonsolvent.

Line 91-92: Nanovesicles were prepared by EI method using optimized production parameters detailed in  previous  works  [13,36,37].

Is it really necessary to refer to three works by the proposing authors for the simple EI procedure? I do not know the MDPI policy regarding self-citations but in the text with a quick glance. Perhaps authors should limit themselves to those that are really necessary.

Line 97-98: “Organic  phase  was  completely  eliminated  using  a  rotary  evaporator  working at 80 mbar in a heating bath at 50ºC”.

Surprising statement: the liposomes prepared by solvent injection process, indeed, are inevitably contaminated by organic solvents, especially ethanol due to formation of azeotrope mixture with water.

Figure 3: panels A and B are not indicated

Table 1: the uncertainties on the values are not reported correctly

Line 206-207: “For both type of formulations tested it can be observed that the determinant step for vesicles  formation by EI method was the injection step, so the optimization of the operation conditions during  the injection is crucial in order to control the vesicle size”.

See comments to Line 69-70. Moreover to circumvent the limitations of the EI technique, researchers have been making significant changes from several years such as the microfluidic injection method. This method is similar to the conventional ethanol injection method, but it enables continuous manipulation of particle size by providing well-defined mixing and a constant water-to-ethanol flow ratio.

Lines 209-211: “The evaporation step, in which the ethanol is eliminated, reduces 11-18% of the sizes respect the  size obtained after the injection (for both types of formulations tested), probably due to the fact that  the membrane components must be more packed to satisfy their lipophilic behaviors once the ethanol  is removed from the system”.

Ethanol cannot be completely removed from the solution with a rotavapor, see comments on line 97-98. Moreover, the possibility of various biologically active macromolecules to inactivation exist in the presence of even low amounts of ethanol.

Lines 219-220: “his observation indicates that sonication is a crucial step for transfersomes formation with EI  method  and  hence  optimization  of  the  operation  parameters  in  that  step  is  necessary  in  order to  control the final nanovesicle size”.

Ethanol  injection  method was  reported  by  Batzri  and  Korn  in  1973  as  one  of the  alternative  methods  for  preparing  SUVs  without sonication  and  regardless  of  drug  solubility. See Single bilayer liposomes prepared without sonication  DOI: 10.1016/0005-2736(73)90408-2. Therefore the EI method was born precisely to avoid sonication. What advantage do the authors find in applying the EI and sonication methods one after the other compared to the sonication method applied directly or after thin film hydration? The sonication method, moreover, is not free from problems such as degradation of the lipid material or of the encapsulated payload and release of metallic material (titanium) from the probe.

Author Response

Reviewer 1

  1. Estupiñan and co-authors in this article titled "Vitamin D3 loaded niosomes and transfersomes produced by ethanol injection method: Identification of the critical preparation step for size control" claim to have identified for the first time "the identification of the crucial step at nanovesicles (transfersomes and niosomes) preparation by Ethanol Injection Method". In my opinion the work does not report elements of novelty with respect to the topic addressed and suffers from theoretical and methodological limitations, mainly the lack of a true experimental design. Detailed comments follow.

Lines 62-63: “The conventional preparation methods are thin film hydration, reverse-phase evaporation and solvent (frequently ethanol) injection and ethanol injection method (EI)”. Another established and fundamental method for vesicle preparation is the “Detergent Depletion”. An interesting and recent application in the field of drug delivery is found in "Preparation of drug-loaded small unilamellar liposomes and evaluation of their potential for the treatment of chronic respiratory diseases", DOI: 10.1016/j.ijpharm.2018.04.030.

We thank the referee for these comments. We have used experimental design in previous studies for the optimization of size tuned nanovesicles preparation (García-Manrique et al., 2016). But in this work we aimed to assist the initial selection of the significant experimental parameters. This has been done by a serie of simple experiments in this work. We believe that the results will be useful for further experimental designs and processes scale-up.

Authors thank the reference valuable suggested. The cited method and the reference is included in the revised version of the manuscript, which enrich the introduction section of the present work.

García-Manrique, P., Matos, M., Gutiérrez, G., Oscar R Estupiñan, Blanco-López, M.C., Pazos, C., Using Factorial Experimental Design to Prepare Size-Tuned Nanovesicles, Industrial and Enginering Chemistry Research, 55 (2016) 9164-9175.

Line 65: “This method allows to obtain small nanovesicles with narrow distribution”. The liposomes prepared by solvent injection process are actually mostly polydispersed.

Authors agree, and it has been corrected in the revised version of the manuscript. Our aim was to highlight that the use of ethanol injection method allows to obtain small unilamellar vesicles instead of monodisperse.

Line 69-70: “but the individual role of each step (injection, evaporation and sonication) on vesicles size has never been reported before”. An extensive literature reports that the critical process parameters that can affect the vesicle size are varied lipids proportion in ethanol, the rate of injection of lipids, magnetic stirring speed, and the phase volume ratio of solvent and nonsolvent.

Authors agree that the individual effect of each parameter has been studied in previous works were solvent injection method has been used for vesicles preparation. However, no other work has been found  in the literature studying the effect of the each individual step (injection, evaporation and sonication) on the global method for vesicle size control. Moreover, the individual effect produced on vesicles of different nature has been scarcely reported. The novelty has been highlighted on the introduction section of the revised version of the manuscript.

Line 91-92: Nanovesicles were prepared by EI method using optimized production parameters detailed in previous works [13,36,37]. Is it really necessary to refer to three works by the proposing authors for the simple EI procedure? I do not know the MDPI policy regarding self-citations but in the text with a quick glance. Perhaps authors should limit themselves to those that are really necessary.

One reference it is enough to describe the used procedure and justify the selected conditions. It has been modified in the revised version of the manuscript, selection the more representative one.

Line 97-98: “Organic phase was completely eliminated using a rotary evaporator working at 80 mbar in a heating bath at 50ºC”. Surprising statement: the liposomes prepared by solvent injection process, indeed, are inevitably contaminated by organic solvents, especially ethanol due to formation of azeotrope mixture with water.

Authors agree that small amount of ethanol could be on the final nanovesicles formed, specially due to the ethanol-water azeotrope. It has been modified in the revised version of the manuscript.

Figure 3: panels A and B are not indicated

Authors thank the notice. Panels has been included in the revised version of the manuscript.

Table 1: the uncertainties on the values are not reported correctly

Uncertainties on the values reported on Table 1 are corrected on the revised version of the manuscript.

Line 206-207: “For both type of formulations tested it can be observed that the determinant step for vesicles formation by EI method was the injection step, so the optimization of the operation conditions during the injection is crucial in order to control the vesicle size”. See comments to Line 69-70. Moreover to circumvent the limitations of the EI technique, researchers have been making significant changes from several years such as the microfluidic injection method. This method is similar to the conventional ethanol injection method, but it enables continuous manipulation of particle size by providing well-defined mixing and a constant water-to-ethanol flow ratio.

Authors agree that ethanol injection method has a large amount of limitations and frequently not enough control on the final vesicle size obtained. Other techniques such as flow-focusing microfluidics or array microcannels, which similar principal to solvent injection method, can also solve this problem, which has been also been explored in our research group. However, these techniques are not yet extensively spread and rarely industrially applied. They have limitations, such as the use of specific instruments which also need to be cleaned and manipulated with special care. They could easily suffer from obstructions or section changes due to material dilatations which will give undesired changes on operational conditions. So, even other techniques can be used with better results, ethanol injection method is one of the most frequently used techniques and one that can be easily scale-up for further industrial applications. This point of view has been included in the introductions section of the revised version of the manuscript.

Lines 209-211: “The evaporation step, in which the ethanol is eliminated, reduces 11-18% of the sizes respect the size obtained after the injection (for both types of formulations tested), probably due to the fact that the membrane components must be more packed to satisfy their lipophilic behaviors once the ethanol is removed from the system”. Ethanol cannot be completely removed from the solution with a rotavapor, see comments on line 97-98. Moreover, the possibility of various biologically active macromolecules to inactivation exist in the presence of even low amounts of ethanol.

Authors agree that the solvent used can not be completely removed by vacuum evaporation, and hence some concentration will remain in the final formulation. Although the presence of ethanol can be a problem on some applications, some advantages have also been found, such as their activity as penetrance enhancer for dermal applications. This point has been included in the revised version of the manuscript.

Lines 219-220: “his observation indicates that sonication is a crucial step for transfersomes formation with EI method and hence optimization of the operation parameters in that step is necessary in order to control the final nanovesicle size”. Ethanol injection method was reported by Batzri and Korn in 1973 as one of the alternative methods for preparing SUVs without sonication and regardless of drug solubility. See Single bilayer liposomes prepared without sonication DOI: 10.1016/0005-2736(73)90408-2. Therefore the EI method was born precisely to avoid sonication. What advantage do the authors find in applying the EI and sonication methods one after the other compared to the sonication method applied directly or after thin film hydration? The sonication method, moreover, is not free from problems such as degradation of the lipid material or of the encapsulated payload and release of metallic material (titanium) from the probe.

Authors agree that sonication step is a really variable step since it is subjected to metal probe degradation during operation. As Batzri and Korn indicate, vesicles are formed without the use of sonication. However sonication is frequently used after ethanol injection and solvent evaporation in order to reduce vesicles size. The use of sonications after a method such as thin film, has been reported to provide different types of vesicles, since they tend to be more multilayer instead of unilamellar. In the present work, we would like to highlight that the conventionally use of sonication is not always necessary. In fact, when niosomes (free of phosphatidylcholine) are prepared, this step can be skipped without significant effect on the final size obtained. This has been highlighted in the conclusions section of the manuscript.

Reviewer 2 Report

  1. Few typographical and grammatical errors were observed.
  2. Elaborate introduction section to highlight the importance and rationale of reported work.
  3. Improvise conclusion section to provide additional details on clinical/commercial significance or  impact of reported results.
  4. Authors should discuss limitations of reported work.
  5.  Provide high resolution images

Author Response

  1. Few typographical and grammatical errors were observed.

The manuscript has been revised and typographical and grammatical errors corrected

  1. Elaborate introduction section to highlight the importance and rationale of reported work.

Introduction section of the revised version of the manuscript has been improved in order to highlight the importance and novelty of the present work

  1. Improvise conclusion section to provide additional details on clinical/commercial significance or impact of reported results.

Conclusion section has been improved in order to highlight its potential application on commercial applications such as food, cosmetic, pharmaceutical and clinical fields.

  1. Authors should discuss limitations of reported work.

Limitation of the reported work has been included in the introduction section of the revised version of the manuscript.

  1. Provide high resolution images

Resolution of the images has been improved in the revised version of the manuscript

Reviewer 3 Report

In the manuscript, authors were sought to develop a nanocarrier encapsulation method that can be applied to drug delivery and food industry. The presented data seems to have a broad interest to researchers in the field of biomedical science and food industry.Authors found that the EI method is crucial step for particle size control and scaling-up production of niosomes and transfersomes. Compared with other published material, the developed method by authors will help to control size of nanovesicles and produce ones at lower cost.

The main points in the text is clear, but the manuscript still needs to be grammatically improved. Authors should carefully check typographical and grammatical errors;

No statistics information and detailed explanation are found in Fig. legends
Line 130, 30 C ???
Line 154-155 should be removed.
Line 158 “is” should be removed.
and other typographical and grammatical errors

Author Response

In the present study, Estupinan et al., developed a method in which EI step was added in the preparation of transfersomes and niosomes, and investigated its potential use in encapsulation of dietary supplement (VitD3). Authors found that transfersomes final size was clearly influenced by the sonication step while niosomes final size was mainly governed by the injection step.

The presented study potentially has impact on the food industry and researchers in the field.

Authors should carefully check typographical and grammatical errors;

  1. No statistics information and detailed explanation are found in Fig. legends

Statistics information is included at the figure captions of the revised version of the manuscript. Moreover, statistics differences on experimental values are mentioned on the results section.

  1. Line 130, 30 C ???

This typing mistake has been corrected (30  ÌŠC). This was the selected temperature in order to study the samples stability, since it is slightly high to the one considered room temperature (25ÌŠ C) and are temperature frequently found in summer time in some indoors. Since at general trend temperature is increasing colloidal system instability to study stability at this temperature was considered adequate to see its behavior at slightly high extreme conditions. This comment is added in the revised version of the manuscript.

  1. Line 154-155 should be removed.

We Agree. It has been eliminated at the version of the manuscript.

  1. Line 158 “is” should be removed.

We agree. It has been removed, from the revised version of the manuscript.

  1. and other typographical and grammatical errors

The manuscript has been revised and typographical and grammatical errors corrected.

Round 2

Reviewer 1 Report

Line 16 and 79: the arguments provided by the authors remain vague and do not change the opinion I expressed previously with respect to the individual role of each step of preparation in EI. So please edit the text by removing "for the first time" or similar expressions.

Line 23: “by” or “for” EI?? Please check.

Line 48: the comma is missing after "biomedicine", Please check.

Line 53: “smaller” or “small”? Please check.

Line 81: please delete  “particle”

Line 104_105: “The injection was made at a constant flow rate of  104130 ml/h”: please explain how.

Line 108: please indicate the power output in watts of the ultrasonic probe.

Table 1:  PDI values have decimal separator with comma, please correct.

Line 191: “ised” Please check.

Figure 3: please indicate which differences are significant.

Line 293 Please check.

Figure 6: panels A and B are not indicated

line 340-341: “Transfersomes  showed  a  precipitation  layer  at  the  bottom  part  of  the  cell  while  in  niosomes  any  precipitation  was  observed”, please be clearer.

Author Response

Journal:Foods

Ms. Ref. No.: foods-915585

Title: Vitamin D3 loaded niosomes and transfersomes produced by ethanol injection method: Identification of the critical preparation step for size control

Authors: Oscar R Estupiñan, Pablo García-Manrique, Maria del Carmen Blanco-López, Maria Matos, Gemma Gutiérrez

We would like to thank the reviewers for their comments on our manuscript entitled Vitamin D3 loaded niosomes and transfersomes produced by ethanol injection method: Identification of the critical preparation step for size control (Ref.:foods-915585). They enquire practical and stimulating questions. After careful revision and taking these comments into consideration, changes in the manuscript have been made, and are highlighted in blue the revised manuscript in blue text. These changes are discussed in the following paragraphs.

Reviewer 1

Line 16 and 79: the arguments provided by the authors remain vague and do not change the opinion I expressed previously with respect to the individual role of each step of preparation in EI. So please edit the text by removing "for the first time" or similar expressions.

It has been revised in the revised version of the manuscript.

Line 23: “by” or “for” EI?? Please check.

It has been corrected in the revised version of the manuscript

Line 48: the comma is missing after "biomedicine", Please check.

A comma has been inserted in the revised version of the manuscript

Line 53: “smaller” or “small”? Please check.

It has been corrected in the revised version of the manuscript

Line 81: please delete “particle”

It has been deleted in the revised version of the manuscript

Line 104_105: “The injection was made at a constant flow rate of104130 ml/h”: please explain how.

Ethanol solution was injected on the aqueous phase by the use of a syringe pump. This information has been included in the revised version of the manuscript.

Line 108: please indicate the power output in watts of the ultrasonic probe.

Ultrasounds probe was working at 900 watts. This information is included in the revised version of the manuscript.

Table 1: PDI values have decimal separator with comma, please correct.

It has been corrected in the revised version of the manuscript.

Line 191: “ised” Please check.

It has been corrected in the revised version of the manuscript

Figure 3: please indicate which differences are significant.

Significant differences (P < 0.05) between size samples have been found after each preparation step for all samples prepared. This information is included in the figure caption of the revised version of the manuscript In some cases, the difference samong samples  were not significant (P<0.05), and this has also been indicated in the revised version of figure 3. Superscripts letters on the figure are included to identify samples without significant size differences on size amongthem.

This information is also included on the manuscript text.

Line 293 Please check.

Sentence has been revised and changed into “Even all values obtained were high, the general trend indicated that the EE was larger on transfersomes than in niosomes”

Figure 6: panels A and B are not indicated

Panels A, B on Figure 6 has been included in the revised version of the manuscript.

line 340-341: “Transfersomes showed a precipitation layer at the bottompart of the cell while in niosomes any precipitation was observed”, please be clearer.

A more detailed explanation has been included in the revised version of the manuscript. “In all transfersomes formulations a precipitation layer was observed (Figure 8), since an increase on backscattering was obtained at the bottom part of the cell.For niosomes this precipitation layer was nearly nonexistent, especially for sample A1 (Figure 7)”
